# Evaluating the Spatial Representativeness of the MODerate Resolution Image Spectroradiometer Albedo Product (MCD43) at AmeriFlux Sites

**Hongmin Zhou** [1,2,*], **Shunlin Liang** [3,4], **Tao He** [4], **Jindi Wang** [1,2], **Yanchen Bo** [1,2] and **Dongdong Wang** [3]

1    The State Key Laboratory of Remote Sensing Science, Institute of Remote Sensing Science and Engineering, Faculty of Geographical Science, Beijing Normal University, Beijing 100875, China; wangjd@bnu.edu.cn (J.W.); boyc@bnu.edu.cn (Y.B.)
2    Beijing Engineering Research Center for Global Land Remote Sensing Products, Beijing Normal University, Beijing 100875, China
3    Department of Geographical Sciences, University of Maryland, College Park, MD 20742, USA; sliang@umd.edu (S.L.); ddwang@umd.edu (D.W.)
4    School of Remote Sensing and Information Engineering, Wuhan University, Wuhan 430079, China; taohers@whu.edu.cn
*    Correspondence: zhouhm@bnu.edu.cn; Tel.: +86-10-5880-6011

**Abstract:** Land surface albedo is a key parameter in regulating surface radiation budgets. The gridded remote sensing albedo product often represents information concerning an area larger than the nominal spatial resolution because of the large viewing angles of the observations. It is essential to quantify the spatial representativeness of remote sensing products to better guide the sampling strategy in field experiments and match products from different sources. This study quantifies the spatial representativeness of the MODerate Resolution Image Spectroradiometer (MODIS) (collection V006) 500 m daily albedo product (MCD43A3) using the high-resolution product as intermediate data for different land cover types. A total of 1820 paired high-resolution Landsat Thematic Mapper (TM) and coarse-resolution (MODIS) albedo data from five land cover types were used. The TM albedo data was used as the spatial-complete high resolution data to evaluate the spatial representativeness of the MODIS albedo product. Semivarioagrams were estimated from 30 m Landsat data at different spatial scales. Surface heterogeneity was evaluated with sill value and relative coefficient of variation. The 30 m Landsat albedo data was aggregated to 450 m–1800 m using two different methods and compared with MODIS albedo product. The spatial representativeness of MODIS albedo product was determined according to the surface heterogeneity and the consistency of MODIS data and the aggregated TM value. Results indicated that for evergreen broadleaf forests, deciduous broadleaf forests, open shrub lands, woody savannas and grasslands, the MODIS 500 m daily albedo product represents a spatial scale of approximately 630 m. For mixed forests and croplands, the representative spatial scale was approximately 690 m. The difference obtained was primarily because of the complexity of the landscape structure. For mixed forests and croplands, the structure of the landscape was relatively complex due to the presence of different forest and plant types in the pixel area, whereas the other landscape structures were considerably simpler.

**Keywords:** albedo; spatial representativeness; semivariogram; MODIS; Landsat

## 1. Introduction

Land surface albedo describes the fraction of incoming solar radiation reflected by the surface of the land. It influences the surface energy budget [1], and it is essential for global and regional estimation of energy and mass exchanges between the Earth's surface and the atmosphere [2–6]. An accuracy of 0.02–0.05 is required by climate models for global surface albedo [7,8]. To monitor the spatio-temporal changes in land surface albedo, albedo products are routinely generated from satellite data, such as the Polarization and Directionality of the Earth's Reflectance (POLDER) [9–12], the Medium Resolution Imaging Spectrometer (MERIS) [13], the Clouds and the Earth's Radiant Energy System (CERES) [11,12], the Visible Infrared Imaging Radiometer Suite (VIIRS) [14,15], and the Airborne Visible Infrared Imaging Spectrometer (AVIRIS) [16]. The Moderate Resolution Imaging Spectroradiometer (MODIS) onboard the Earth Observation System (EOS) Terra and Aqua satellites routinely provided data to derive the land surface shortwave and visible albedos [17–19] used to calibrate and improve albedo parameterizations for land, weather, and climate models [20–26]. Assessment of the accuracy of these products is important because it is critical to the scientific community for various applications. Feedback from this activity will help improve the generation of these products [27].

Direct comparison with the ground-based observations of albedo values is the commonly used method to assess the accuracy of remote sensing albedo products [28–30]. The tower observation is compared directly with the satellite product [31–33] based on the assumption that the satellite and tower observation have the same footprint or the landscape is homogeneous. Researchers subsequently recognized that the land surface albedo varies strongly in space and across seasons, because of which land surface homogeneity is now examined prior to the "point-to-pixel" comparison. Only sites whose representativeness is adequate for satellite pixel scales are selected in the direct validation and "heterogeneous" sites are excluded [34]. The method most suited to a comparison for all kinds of landscapes is using higher spatial resolution data as intermediate data [27,35,36]. Burakowski, et al. [37] used airborne hyperspectral imagery to validate MODIS albedo product in snow-covered areas; Mira, et al. [10] used the convolved albedo onboard the Formosat-2 Taiwanese satellite as a reference to evaluate the newly released MCD43D product. However, since no global high-resolution albedo product (at a level of tens of meters) is available, validation using intermediate data has been conducted at a limited number of locations.

Prior to evaluating the remote sensing product, identifying the spatial representativeness of the products is essential. It can help ground sampling point settings and match remote sensing data from multiple sources. MODIS albedo products are retrieved using observations covering a large area that depends on the view zenith angles. Although observations are weighted by angular coverage before albedo retrieval, the actual coverage of the pixel is always larger than the nominal spatial resolution [38,39]. Efforts have been made to characterize the effective resolution of the MODIS gridded product [40]. Mira, et al. [10] used Formost-2 data at a resolution of 8 m to characterize the equivalent point spread function of MODIS albedo at a 1 km pixel. The Full Width at Half Maximum (FWHM), recognized as the effective resolution, has been confirmed to represent the footprint of MODIS data for accurate validation. Campagnolo et al. [41] used extensive time series data (2003–2014) at a large size of the linear natural target in the Netherlands to analyze the effective spatial resolution of the MODIS albedo product (the spatial scale the pixel represents). They verified that the spatial representativeness of the MODIS daily albedo product approximately varied from 606 m to 843 m. Campagnolo and Montano [40] estimated the point spread function (PSF) of nominal 250 m MODIS gridded surface reflectance products, and discovered that the spatial representativeness varied from 344 m to 835 m along the rows, and between 292 m and 523 m along the columns. Their work helps users understand the spatial properties of the satellite product, but their work was based only on a single area, and a general adaptable result that is based on different kinds of land surfaces is needed.

The MODIS BRDF/albedo product was derived with a semi-empirical, kernel-driven BRDF model. Data for 16-day, multi-angular, cloud-free, atmosphere-corrected surface reflectance was compiled to apply the retrieval procedure. To better characterize the rapid change of the land surface,

the daily albedo product was retrieved using the same semi-empirical algorithm, but with the single day of interest emphasized by being weighted more heavily [39,42] in the 16-day composite period. The spatial representativeness and accuracy of the newly released daily albedo product (MCD43A3, V006) have not been tested for different types of land cover for a long time series.

The high-resolution (30–80 m) satellite albedo product is essential in understanding the climatic consequences of land cover change and medium-to-fine scale applications [43]. It is also a key bridge to the assessment of coarse-resolution products. He et al. [44] developed a method to estimate both snow-covered and snow-free albedo from the Chinese environment and disaster monitoring and forecasting small satellite constellation (HJ) satellite data. Zhou, et al. [15] then derived 30 m albedo from Landsat 7 and Landsat 8 using a similar algorithm. He et al. [45] evaluated 30 m albedo product estimated from Landsat Multispectral Scanner (MSS), Thematic Mapper (TM), Enhanced Thematic Mapper Plus (ETM+), and Operational Land Imager (OLI) at Surface Radiation (SURFRAD), AmeriFlux, Baseline Surface Radiation Network (BSRN), and Greenland Climate Network (GC-Net) sites, with results indicating that the direct estimation approach can generate reliable albedo estimates with accuracy of 0.022 to 0.034 in terms of the root mean square error (RMSE). The derivation of global land surface albedo product using Landsat sensors makes it possible to better understand energy transfer between the land surface and the atmosphere at global and regional scales. Furthermore, it makes the assessment of the coarse-resolution albedo product possible, as well as scale transformations for different land cover types and landscapes.

## 2. Datasets

Three types of datasets were used to quantitatively determine the spatial representativeness of the MODIS daily albedo product: (1) Tower-measured surface albedo datasets from AmeriFlux sites were used as field truth to calibrate 30 m albedo data. (2) The MODIS daily albedo product MOD43A3 (V006, 500 m, daily) was assessed, and annual land cover data were downloaded to identify different land cover types. (3) Landsat TM level-one data were used to estimate the 30 m spatial resolution of the albedo, which was then used as the spatial complete high resolution data to evaluate the spatial representativeness of the MODIS 500 m spatial resolution albedo product.

### 2.1. AmeriFlux Site Data

The AmeriFlux network is a community of sites and scientists measuring the amounts of carbon, water, energy fluxes, and related environmental variables in ecosystems across the Americas [46]. It supplies a long period of field observations and features wide coverage of forests, grasslands, croplands, shrub lands, wetlands, savannas, and other geographies (e.g., urban). A total of 166 sites are distributed in North and South America, of which 109 feature continuous radiation measurement. Level-2 data of downward and upward radiation from these 109 sites were downloaded from the Carbon Dioxide Information Analysis Center (CDIAC) at Oak Ridge National Laboratory (ORNL). The instruments mounted on the tower including Kipp and Zonen (CNR1, CM-3, OR CM-6), and the Eppley-PSP albedometer/pyranometer. The instruments were fitted with domes to collect radiation fluxes in the broadband shortwave domain (0.3–2.8 μm). Data received from each site were reviewed and incorporated into the network of the AmeriFlux database. The data review process includes checking for consistent units, naming conventions, reporting intervals, and reforming to ensure consistency with the larger network-wide database [47]. Radiation data were collected every half hour during the years indicated in Table 1 for each site. The overall range of years recorded at one site or more is 1995 to 2013. A region of 10 × 10 km around each site was selected as the research area. Information pertaining to the sites used in this study—site name, latitude, longitude, land cover type, and the years when data are available—is listed in Table 1.

**Table 1.** Flux sites used in the analysis.

| SITE_NAME | LOCATION_LAT | LOCATION_LONG | IGBP | TOWER_BEGAN | TOWER_END | SITE_NAME | LOCATION_LAT | LOCATION_LONG | IGBP | TOWER_BEGAN | TOWER_END |
|---|---|---|---|---|---|---|---|---|---|---|---|
| ARM Southern Great Plains site—Lamont | 36.61 | −97.49 | CRO | 2002 | 2013 | Mary's River (Fir) site | 44.65 | −123.55 | ENF | 2005 | 2013 |
| Bondville | 40.01 | −88.29 | CRO | 1996 | 2013 | Metolius Young Pine Burn | 44.32 | −121.61 | ENF | 2010 | 2013 |
| Bondville (companion site) | 40.01 | −88.29 | CRO | 2004 | 2008 | Flagstaff—Unmanaged Forest | 35.09 | −111.76 | ENF | 2006 | 2010 |
| Brooks Field Site 10—Ames | 41.69 | −93.69 | CRO | 2001 | 2013 | Duke Forest—loblolly pine | 35.98 | −79.09 | ENF | 2001 | 2008 |
| Brooks Field Site 11—Ames | 41.97 | −93.69 | CRO | 2001 | 2013 | Howland Forest (west tower) | 45.21 | −68.75 | ENF | 1999 | 2013 |
| Curtice Walter—Berger cropland | 41.63 | −83.35 | CRO | 2011 | 2013 | Metolius—second young aged pine | 44.32 | −121.61 | ENF | 2004 | 2009 |
| Fermi National Accelerator Laboratory—Batavia (Agricultural site) | 41.86 | −88.22 | CRO | 2005 | 2013 | Metolius—intermediate aged ponderosa pine | 44.45 | −121.56 | ENF | 2002 | 2013 |
| Mead—irrigated continuous maize site | 41.17 | −96.48 | CRO | 2001 | 2013 | Howland Forest (main tower) | 45.20 | −68.74 | ENF | 1996 | 2013 |
| Mead—irrigated maize-soybean rotation site | 41.16 | −96.47 | CRO | 2001 | 2013 | Poker Flat Research Range Black Spruce Forest | 65.12 | −147.49 | ENF | 2011 | 2013 |
| Mead—rainfed maize-soybean rotation site | 41.18 | −96.44 | CRO | 2001 | 2013 | Saskatchewan—Western Boreal, forest burned in 1977. | 54.49 | −105.82 | ENF | 2004 | 2006 |
| Ponca City | 36.77 | −97.13 | CRO | 1997 | 2001 | Flagstaff—Managed Forest | 35.14 | −111.73 | ENF | 2006 | 2010 |
| Rosemount—G21 | 44.43 | −93.05 | CRO | 2002 | 2013 | Niwot Ridge Forest (LTER NWT1) | 40.03 | −105.55 | ENF | 1998 | 2013 |
| Sioux Falls Portable | 43.24 | −96.90 | CRO | 2007 | 2013 | Howland Forest (harvest site) | 45.21 | −68.73 | ENF | 2000 | 2013 |
| Twitchell Corn | 38.10 | −121.64 | CRO | 2012 | 2013 | UCI-1930 burn site | 55.91 | −98.52 | ENF | 2001 | 2005 |
| Twitchell Alfalfa | 38.12 | −121.65 | CRO | 2013 | 2013 | UCI-1964 burn site wet | 55.91 | −98.38 | ENF | 2002 | 2004 |
| Bartlett Experimental Forest | 44.06 | −71.29 | DBF | 2004 | 2013 | Quebec—Eastern Boreal | 49.27 | −74.04 | ENF | 2001 | 2013 |
| Morgan Monroe State Forest | 39.32 | −86.41 | DBF | 1999 | 2013 | NC_Clearcut | 35.81 | −76.71 | ENF | 2005 | 2013 |
| Duke Forest-hardwoods | 35.97 | −79.10 | DBF | 2003 | 2013 | GLEES | 41.37 | −106.24 | ENF | 2004 | 2013 |
| Willow Creek | 45.81 | −90.08 | DBF | 1999 | 2013 | Wind River Crane Site | 45.82 | −121.95 | ENF | 1998 | 2013 |
| Chestnut Ridge | 35.93 | −84.33 | DBF | 2005 | 2013 | UCI-1850 burn site | 55.88 | −98.48 | ENF | 2002 | 2005 |
| Silas Little—New Jersey | 39.91 | −74.60 | DBF | 2004 | 2013 | UCI-1964 burn site | 55.91 | −98.38 | ENF | 2001 | 2005 |
| Univ. of Mich. Biological Station | 45.56 | −84.71 | DBF | 1999 | 2013 | UCI-1981 burn site | 55.86 | −98.49 | ENF | 2001 | 2005 |
| Walker Branch Watershed | 35.96 | −84.29 | DBF | 1995 | 1999 | Black Hills | 44.16 | −103.65 | ENF | 2003 | 2008 |
| Missouri Ozark Site | 38.74 | −92.20 | DBF | 2004 | 2013 | Chimney Park | 41.07 | −106.12 | ENF | 2009 | 2013 |
| Oak Openings | 41.55 | −83.84 | DBF | 2004 | 2013 | NC_Loblolly Plantation | 35.80 | −76.67 | ENF | 2005 | 2013 |
| UMBS Disturbance | 45.56 | −84.70 | DBF | 2007 | 2013 | Quebec—Eastern Boreal, Mature Black Spruce. | 49.69 | −74.34 | ENF | 2003 | 2013 |
| Fermi National Accelerator Laboratory—Batavia | 41.84 | −88.24 | GRA | 2004 | 2013 | Valles Caldera National Preserve (Mixed Conifer) | 35.89 | −106.53 | ENF | 2007 | 2013 |
| ARM USDA UNL OSU Woodward Switchgrass 1 | 36.43 | −99.42 | GRA | 2009 | 2013 | Valles Caldera National Preserve (Ponderosa pine) | 35.86 | −106.60 | ENF | 2007 | 2013 |
| ARM USDA UNL OSU Woodward Switchgrass 2 | 36.64 | −99.60 | GRA | 2009 | 2013 | Ontario—Turkey Point 1939 Plantation White Pine | 42.71 | −80.36 | ENF | 2003 | 2013 |
| Konza Prairie LTER (KNZ) | 39.08 | −96.56 | GRA | 2006 | 2013 | Bonanza Creek | 63.92 | −145.74 | OSH | 2003 | 2013 |
| Walnut Gulch Kendall Grasslands | 31.74 | −109.94 | GRA | 2004 | 2013 | UCI-1989 burn site | 55.92 | −98.96 | OSH | 2001 | 2005 |
| Audubon Research Ranch | 31.59 | −110.51 | GRA | 2002 | 2013 | Saskatchewan—Western Boreal | 54.09 | −106.01 | OSH | 2001 | 2006 |

**Table 1.** *Cont.*

| SITE_NAME | LOCATION_LAT | LOCATION_LONG | IGBP | TOWER_BEGAN | TOWER_END | SITE_NAME | LOCATION_LAT | LOCATION_LONG | IGBP | TOWER_BEGAN | TOWER_END |
|---|---|---|---|---|---|---|---|---|---|---|---|
| Duke Forest-open field | 35.97 | −79.09 | GRA | 2001 | 2013 | Anaktuvuk River Severe Burn | 68.99 | −150.28 | OSH | 2008 | 2013 |
| Diablo | 37.68 | −121.53 | GRA | 2010 | 2013 | Anaktuvuk River Moderate Burn | 68.95 | −150.21 | OSH | 2008 | 2013 |
| Santa Rita Grassland | 31.79 | −110.83 | GRA | 2008 | 2013 | Anaktuvuk River Unburned | 68.93 | −150.27 | OSH | 2008 | 2013 |
| Canaan Valley | 39.06 | −79.42 | GRA | 2004 | 2013 | Walden | 40.78 | −106.26 | OSH | 2006 | 2008 |
| Cottonwood | 43.95 | −101.85 | GRA | 2007 | 2009 | UCI-1998 burn site | 56.64 | −99.95 | OSH | 2002 | 2005 |
| Fort Peck | 48.31 | −105.10 | GRA | 2000 | 2013 | Sevilleta (LTER desert shrubland) | 34.33 | −106.74 | OSH | 2007 | 2013 |
| Brookings | 44.35 | −96.84 | GRA | 2004 | 2013 | Santa Rita Creosote | 31.91 | −110.84 | OSH | 2008 | 2013 |
| Walnut River Watershed (Smileyburg) | 37.52 | −96.86 | GRA | 2001 | 2004 | Juniper savanna site (Willard) | 34.43 | −105.86 | OSH | 2007 | 2013 |
| Sevilleta (LTER desert grassland) | 34.36 | −106.70 | GRA | 2007 | 2013 | Walnut Gulch Lucky Hills Shrub | 31.74 | −110.05 | OSH | 2007 | 2013 |
| Flagstaff—Wildfire | 35.45 | −111.77 | GRA | 2006 | 2010 | Imnavait Creek Watershed Tussock Tundra | 68.61 | −149.30 | OSH | 2007 | 2013 |
| KUOM Turfgrass Field | 45.00 | −93.19 | GRA | 2005 | 2009 | Imnavait Creek Watershed Heath Tundra | 68.61 | −149.30 | OSH | 2007 | 2013 |
| Kansas Field Station | 39.06 | −95.19 | GRA | 2007 | 2013 | Pinon-juniper site (Mountainair) | 34.44 | −106.24 | OSH | 2007 | 2013 |
| Corral Pocket | 38.09 | −109.39 | GRA | 2001 | 2007 | Everglades (long hydroperiod marsh) | 25.55 | −80.78 | WET | 2007 | 2013 |
| Shidler—Oklahoma | 36.93 | −96.68 | GRA | 1997 | 2001 | Everglades (short hydroperiod marsh) | 25.44 | −80.59 | WET | 2007 | 2013 |
| Vaira Ranch—Ione | 38.41 | −120.95 | GRA | 2000 | 2013 | Twitchell East End Wetland | 38.10 | −121.64 | WET | 2013 | 2013 |
| Goodwin Creek | 34.25 | −89.87 | GRA | 2002 | 2006 | Winous Point North Marsh | 41.46 | −83.00 | WET | 2011 | 2013 |
| Santarem-Km83-Logged Forest | -3.02 | −54.97 | EBF | 2000 | 2003 | Twitchell Wetland West Pond | 38.11 | −121.65 | WET | 2012 | 2013 |
| Shark River Slough (Tower SRS-6) Everglades | 25.36 | −81.08 | EBF | 2004 | 2013 | Imnavait Creek Watershed Wet Sedge Tundra | 68.61 | −149.31 | WET | 2007 | 2013 |
| Ontario—Groundhog River, Boreal Mixedwood Forest. | 48.22 | −82.16 | MF | 2003 | 2013 | Olentangy River Wetland Research Park | 40.02 | −83.02 | WET | 2011 | 2013 |
| Santa Rita Mesquite | 31.82 | −110.87 | WSA | 2004 | 2013 | Ivotuk | 68.49 | −155.75 | WET | 2003 | 2013 |
| Freeman Ranch—Mesquite Juniper | 29.95 | −98.00 | WSA | 2004 | 2013 | Lost Creek | 46.08 | −89.98 | WET | 2001 | 2013 |
| Sylvania Wilderness Area | 46.24 | −89.35 | MF | 2001 | 2013 | Freeman Ranch—Woodland | 29.94 | −97.99 | CSH | 2004 | 2013 |
| Fort Dix | 39.97 | −74.43 | MF | 2005 | 2008 | | | | | | |

Note: * SITE_NAME is the full name of the selected AmeriFlux sites. *IGBP* is the International Geosphere Biosphere Program.CRO is cropland, DBF is deciduous broadleaf forest, ENF is evergreen needleleaf forest, GRA is grassland, MF is mixed forest, OSH is open shrubs, WET is permanent wetlands, WSA is woody savannas.

### 2.2. MODIS Data

#### 2.2.1. MCD43 BRDF/Albedo Product

The MODIS Albedo product (MCD43A3 V06) provides daily albedo data on the Earth's surface for each pixel at a grid resolution of 500 m. A 16-day period of cloud-free surface reflectance from both the Terra and the Aqua is used to derive the daily data, with weight as a function of quality, the observation coverage, and the temporal distance from the day of interest. The newly broadcasted albedo product has been utilized for regional applications (e.g., forest, agriculture, and disturbance monitoring), and is now downloadable from the Land Processes Distributed Active Archive Center (LP DAAC). The 16-day composed daily albedo product has been validated over typical agricultural landscapes [10,41], grasslands, and agriculture and forest surfaces [39], but has not been extensively validated for other land cover types using massive amounts of data.

#### 2.2.2. MCD12Q1 Land Cover

Land cover plays a major role in the climate and biogeochemistry of the Earth's ecosystem. The MODIS land cover type product provides a suite of land cover types by using spectral and temporal information derived from the MODIS. It characterizes five global land cover classification systems—the International Geosphere Biosphere Program (*IGBP*), university of Maryland (UMD), Leaf Area Index/Fraction of Photosynthetically Active Radiation absorbed by vegetation (LAI/fPAR), New Patriotic Party (NPP), and Plant Functional Types (PFT)—at annual time steps and a 500 m spatial resolution. Land cover type assessment and quality control information are also included. Given that the land cover type may change over time, yearly MODIS land cover type data (MCD12Q1) are downloaded, and the IGBP land cover type is used.

### 2.3. Landsat-Retrieved Albedo

The TM onboard the Landsat 4 and 5 satellites with seven spectral bands covered the shortwave range at a resolution of 30 m from 1984 to 2011. It is an optimal data source for regional and global land surface change research because of its long period of operation [48–52]. The high-spatial-resolution (30 m), long-term albedo product based on Landsat data has been derived [43,45,53]. In this study, we adopt the algorithm of He et al. [45] to derive the global long-term land surface albedo product from Landsat TM data. L1T data that were temporally consistent with AmeriFlux site observations were downloaded from the U.S. Geological Survey (USGS) website. To minimize the influence of cloud coverage, only the highest data quality (quality flag is 9) and a maximum cloud cover of 10% were used. In this paper, 2034 scenes of TM data were downloaded, covering all land cover types and seasons, where 1820 scenes corresponded to high-quality MODIS data. TM data were first rectified to sinusoidal projections and resampled to 30 m resolution using the nearest neighbor algorithm.

The accuracy of meso-scale data needs to be verified and calibrated before they are used to bridge field measurements and coarse-resolution products. Landsat TM albedo data was first calibrated with the field data. The Landsat TM overpassed at local time 10:00–11:00 AM. To guarantee the temporal consistency between TM data and field observation, field observations 15 min before and after the TM passing over were averaged. The field observation covers an area larger than 30 m, so we assume that the land surface at 30 m scale is homogeneous and field observation can compare directly with TM data. However, for each AmeriFlux site, the tower-located TM pixel was picked out according to the tower location. Sometimes, the tower was not located at the center of the pixel and possible geometric correction error may also exist, so albedo of 3 × 3 TM pixels around the tower were averaged and compared with the field observations.

Figure 1 shows a scatter plot of the field measurements and the TM albedo. The correction coefficient was derived from least squares regression between TM data and ground measurement. The determination coefficient was 0.877, RMSE and bias were 0.033 and 0.00035, respectively. The calibrated

TM data with the determined correlation were then used as a bridge for comparison with the MODIS albedo product as well as for further analysis to determine the spatial representativeness of the later.

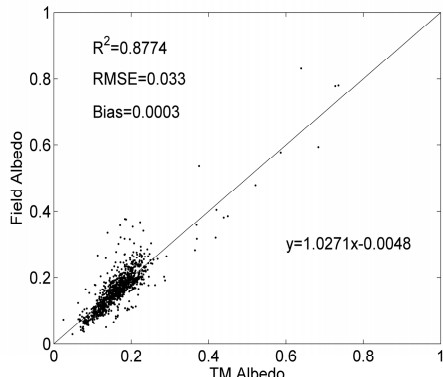

**Figure 1.** Field data comparison with Landsat Thematic Mapper (TM) data, which is the average of $3 \times 3$ pixels. The fitting function is then used for TM albedo data calibration.

## 3. Methods

To quantitatively evaluate the spatial representativeness of the MODIS daily albedo product, 30 m albedo data were derived from Landsat TM data and then calibrated using field observation data. Semivariagrams are calculated with calibrated 30 m albedo data for different scales, in which sill value is considered as the key index to measure the magnitude of field homogeneity, i.e., sill values of adjacent scales differ from each other significantly, indicating that the field homogeneity changed significantly between these two scales. Relative coefficient of variation ($R_{CV}$) is also employed to determine the land surface homogeneity: $R_{CV}$ tends to be 0, which indicates that the adjacent scales have similar homogeneity, the MODIS pixel represents the larger scale. The last index used is the determination coefficient, 30 m albedo data is aggregated to different scales and compared with MODIS data. The scale in which it has the highest determination coefficient indicates the MODIS pixel represents the spatial scale best.

### 3.1. Variogram Model Parameters from TM Data

When using tower observation to validate the remote sensing albedo product, the spatial representativeness of the observation footprint was investigated on semivariogram models [37,39,47,54]. In this study, the method of deriving variogram functions to analyze surface albedo with TM data was used [36]. The variogram estimator $\gamma(h)$ was used to obtain the half-average squared difference between albedo values within a certain distance. According to Román et al. [47], the isotropic spherical variogram model [55] was used to fit the variogram model parameters—the range (a), the sill (c), and the nugget effect ($c_0$), as below:

$$\gamma_{sph}(h) = \begin{cases} c_0 + c\left(1.5\frac{h}{a} - 0.5\left(\frac{h}{a}\right)^3\right) for\ 0 \leq h \leq a \\ c_0 + c\ \ for\ h > a \end{cases} \tag{1}$$

The range is the distance from a point beyond which there is no further correlation of the albedo associated with that point. It is the average patch size of the landscape in landscape ecology, which represents a region that differs from its surroundings, but is not necessarily inter-homogeneous. The sill is the maximum semivariance, and is the ordinate value of the range at which the variogram levels off to an asymptote. The non-zero value of the variogram when h = 0 is called the nugget effect. It depends on the variance associated with small-scale variation, measurement errors, or their combination [56]. The range, the sill, and the nugget effect all reveal the spatial variation of the land surface and the scale effect associated with remote sensing data [47,57]. It has been suggested that

the land surface is homogeneous (representative) when the sill value is less than 0.001 [37]. In this paper, the semivariogram model was used as well. The 30 m Landsat albedo was first re-projected to a sinusoidal projection and the Landsat pixel located at the center of the MODIS 500 m pixel was determined. The semivariogram was calculated from Landsat data. The model parameters were fitted according to the spherical model.

### 3.2. Relative Coefficient of Variation

The indices deduced from the parameters of the semivariogram model and the statistical values were also used in this study. The relative coefficient of variation ($R_{CV}$), the scale requirement index, the relative proportion of structural variation, and the relative strength of spatial correlation, derived from the semivariogram model, were first used by Román et al. [47] to depict spatial variation. When the measurement site was spatially representative, the overall variation between the internal components of the measurement site (scale 1) and the adjacent landscape (scale 2) should have been similar in magnitude, and the $R_{CV}$ should have approached zero. The $R_{CV}$ was also calculated to check the spatial variation in the landscape. To calculate it, the coefficient of variation (cv) was first calculated as the ratio of the standard deviation to the mean. The $R_{CV}$ is given below:

$$R_{CV} = \frac{CV_{scale2} - CV_{scale1}}{CV_{scale1}} \tag{2}$$

### 3.3. Evaluation Strategies

To determine the spatial representativeness of the 500 m product used in this study, variogram estimation was performed at nine scales for each site. When estimating the variogram, the common spatial step was one MODIS pixel, and according to the result of [41], three scales were added between $1 \times 1$ and $2 \times 2$ MODIS pixels; that is, $21 \times 21$, $23 \times 23$, and $29 \times 29$ TM pixels. The $2.5 \times 2.5$ MODIS pixels were also added to keep an intensive estimation. The estimating scales used in this study are shown in Table 2.

**Table 2.** Variogram estimating scales selected in this study.

| Scale | 1 | 2 | 3 | 4 | 5 | 6 | 7 | 8 | 9 |
|---|---|---|---|---|---|---|---|---|---|
| TM pixels | $15 \times 15$ | $21 \times 21$ | $23 \times 23$ | $29 \times 29$ | $31 \times 31$ | $39 \times 39$ | $47 \times 47$ | $61 \times 61$ | $75 \times 75$ |
| Scale in meters | 450 | 630 | 690 | 870 | 930 | 1170 | 1410 | 1830 | 2250 |
| MODIS pixels | $1 \times 1$ | - | - | - | $2 \times 2$ | $2.5 \times 2.5$ | $3 \times 3$ | $4 \times 4$ | $5 \times 5$ |

To make the analysis representative, for each site, not only was the tower-located MODIS pixel analyzed, the research area was enlarged to $9 \times 9$ pixels with the tower-located MODIS pixel as the central pixel. In this research area, for every MODIS pixel, the semivariogram parameters (nugget, sill, range) and the statistical value, including mean and standard deviation, were calculated at the nine scales illustrated in Table 2.

The spatial representativeness was evaluated according to the calculated parameters and values. The sill value represents the magnitude of spatial variability. In this study, the sill values of $9 \times 9$ MODIS pixel in every research area were compiled (for every scale, sill values were compiled as a group; thus, we obtained nine groups of data), and the paired t-test was implemented every adjacent scale data pair (e.g., $15 \times 15$ and $21 \times 21$) to find whether the land surface at adjacent scales was significantly different. Statistical significance was determined by the *p*-value. If the *p*-value was zero, it indicated that the difference of the land surface heterogeneity at adjacent scales was not significant; the central MODIS pixel was able to represent an area determined by the larger scale.

$R_{CV}$ depicts spatial variation in the landscape. In this study, $R_{CV}$ was calculated between each pair of scales in Table 2, then all the data from 109 sites and 81 pixels were compiled and the histogram was plotted. The scale in which the $R_{CV}$ value is smallest indicates that the central pixel has the similar spatial representativeness in these adjacent scales.

Taking the calibrated TM data as the albedo reference, we aggregated the TM value at different scales and compared the aggregated 30 m albedo data directly with the 500 m albedo product, and two aggregation methods were used. One was simple average, and the other considered the point spread function. Campagnolo and Montano [40] and Mira et al. [10] used the convolution of a Gaussian function to characterize the optical PSF of the MODIS instrument, assuming that the central area of the pixel made a greater contribution to the signal. In this paper, we adopt the Gaussian function, as was done by Campagnolo and Montano [40] and Mira et al. [10], but set an asymmetric Gaussian point spread function. The PSF was defined below.

$$\mathrm{PSF}(x, y) = \frac{1}{2\pi a^2} exp^{-0.5(x^2/a^2 + y^2/a^2)} \tag{3}$$

where

$$\mathrm{a} = \frac{\mathrm{FWHM}}{2.355} \tag{4}$$

The FWHM is the full width at half maximum of the PSF. In this study, we set it as the sensor spatial resolution as Campagnolo and Montano [40], which also represents the spatial representativeness of the pixel.

The determination coefficient was used to judge the satisfactoriness of the consistency between the two datasets. The scale where the highest determination coefficient appeared was considered that the MODIS data has the best spatial representativeness.

## 4. Results

### 4.1. Spatial Representativeness Determined by Sill Value

A paired t-test was performed on the sill value for each set of paired TM and MODIS data. The spatial variation was determined according to the criterion that if the *p*-value was zero, it confirmed the null hypothesis—land surface variation at adjacent scales was not different, and the pixel can represent the larger scale scape. For all data pairs, the histogram of spatial variation is shown in Figure 2. The median value was also used as the indicator of effective spatial resolution, as in Campagnolo et al. [41]. In this situation, the median value was 2, which suggests that the variation in the land surface was subtle within the 21 × 21 TM scale, and the proper aggregation scale for TM albedo data was 21 × 21 pixels. The 21 × 21 aggregation scale represented a 630 m spatial scale, which is consistent with the result of [41].

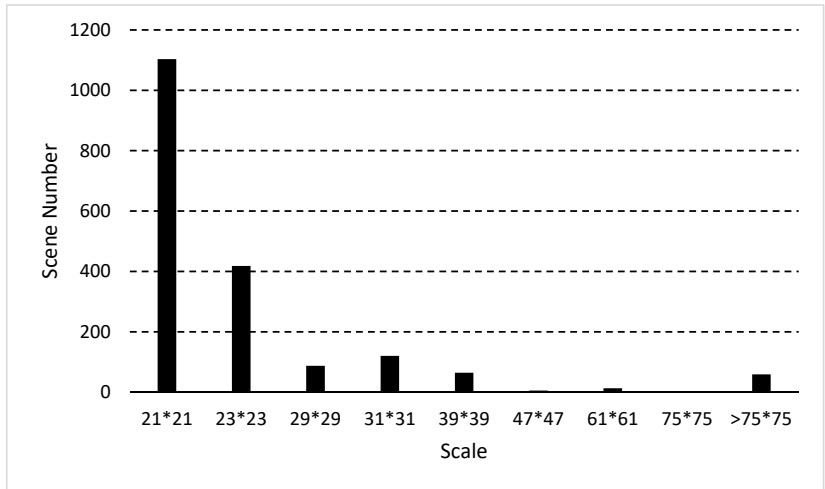

**Figure 2.** The proper aggregation scale histogram according to the sill value of each scene. The scale is defined with TM pixels. The scene number is the total number of the scenes at each scale where the MODerate Resolution Image Spectroradiometer (MODIS) pixel represents the spatial scale well.

Table 3 shows the proper aggregation scale for every land cover type of AmeriFlux sites. For most land cover types, the land surface was stable within 21 × 21 TM pixels, and for mixed forest and croplands, the proposed effective spatial resolution was 690 m (approximately 23 × 23 TM pixels).

**Table 3.** Spatial representativeness of MODIS 500 m albedo product for different land cover types.

| Land Cover Type | Suggested Spatial Scale | Suggested Spatial Representativeness in Meters |
|---|---|---|
| Evergreen Broadleaf forest | 2 | 630 |
| Deciduous Broadleaf forest | 2 | 630 |
| Mixed forest | 3 | 690 |
| Open shrublands | 2 | 630 |
| Woody savannas | 2 | 630 |
| Grasslands | 2 | 630 |
| Croplands | 3 | 690 |

### 4.2. Spatial Representativeness Determined by Relative Coefficient of Variation

Figure 3 shows the histogram of the $R_{CV}$ at each scale. To depict the spatial variation at adjacent scales, the $R_{CV}$ was computed from the CV values at each adjacent scale. The median value was calculated as in Figure 3 to describe the integral spatial variation at each scale. From Figure 3, we can see that $R_{CV1}$, which represents the $R_{CV}$ of the first scale, was the largest for all scales. This means that the spatial variation between scale 1 (15 × 15 TM pixels) and scale 2 (21 × 21 TM pixels) was significant. The median values of $R_{CV2}$ and $R_{CV4}$ were small, which means that the effective spatial resolution should have been 630 m or 870 m. The conclusion was partly consistent with that of the t-test on the sill values. However, the step sizes of the scales were different: for scales 1 and 2, the step was 6 TM pixels, but for scales 2 and 3, it was only 2. Hence, the low level of the variation in the sill value and the small value of the $R_{CV}$ could have been deduced from the small step size. To verify this conclusion, we then reduced the analysis step size and aggregated the TM data at different scales to explore the proper representative scale.

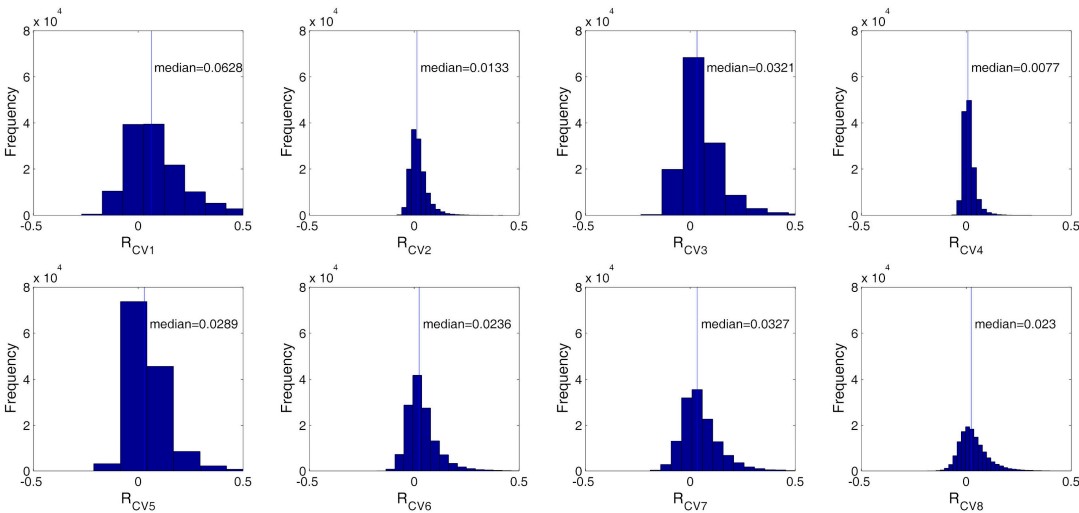

**Figure 3.** Histogram of $R_{CV}$ at (**a**)–(**h**): scale 1–scale 8. The median is the median value of $R_{CV}$ at each scale.

### 4.3. Spatial Representativeness Determined by Direct Comparison

We refined the step size to find the highest correlation coefficient between TM and MODIS to determine the proper aggregation scale and use it as the representative spatial scale of MODIS data. The step size was set to 2 TM pixels. Figure 4 shows that the coefficient varied with aggregation scale. TM data were aggregated from 15 × 15 to 61 × 61 TM pixels. From Figure 4 we can see that when

$23 \times 23$ TM pixels were aggregated, the correlation coefficient was the highest and the root mean squared error was the lowest compared with MODIS data. This corresponds to the results in Table 3 in some degree.

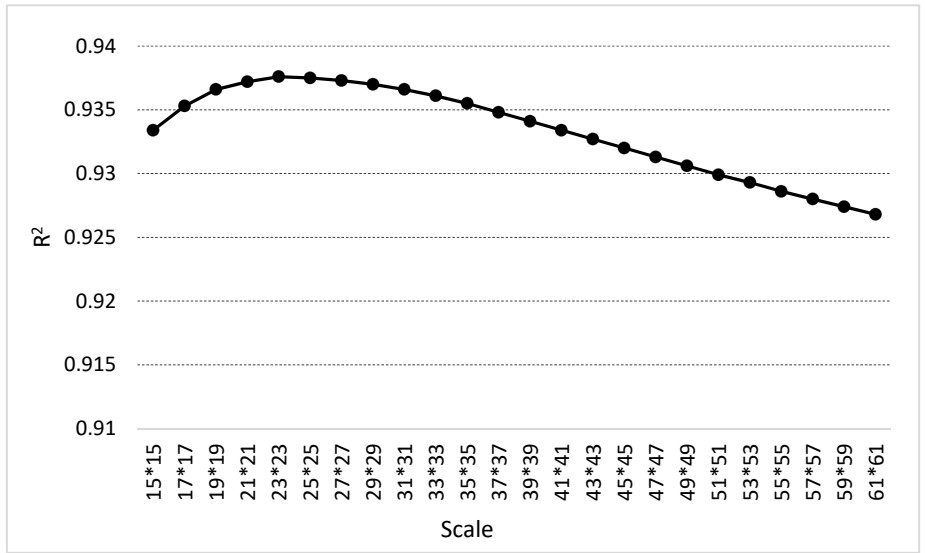

**Figure 4.** $R^2$ variation with aggregation scale.

We then checked the comparative accuracy of different land cover types. Table 4 shows the accuracy of comparison in different land cover types at the suggested aggregation scale according to Table 3. For all the land cover types, the RMSE was less than 0.03 and bias less than 0.018. The $R^2$ of evergreen broadleaf forests was the lowest, mainly because the land surface of this type was snow free, and all albedo values clustered together. For all other land cover types, the $R^2$ value was higher than 0.86, and for croplands, it reaches 0.965.

**Table 4.** Comparison in terms of accuracy between TM and MODIS for different land types.

| LC | Land Cover Type | $R^2$ | RMSE | BIAS |
|----|-----------------|-------|------|------|
| 2 | Evergreen Broadleaf forest | 0.688 | 0.016 | 0.014 |
| 4 | Deciduous Broadleaf forest | 0.863 | 0.018 | 0.013 |
| 5 | Mixed forest | 0.926 | 0.026 | 0.014 |
| 7 | Open shrublands | 0.947 | 0.018 | 0.014 |
| 8 | Woody savannas | 0.944 | 0.030 | 0.017 |
| 10 | Grasslands | 0.941 | 0.021 | 0.009 |
| 12 | Croplands | 0.965 | 0.023 | 0.003 |

## 5. Discussion

### 5.1. Errors Induced by Landsat Albedo Estimation and Spatial Discrepancy

The Landsat albedo estimation algorithm has been validated on a variety of sensors and land cover types [15,16,44,58]. In this study, it was first calibrated using field observations. The errors induced by the estimation algorithm were hence eliminated.

In most cases, the sites were not located at the center of a MODIS pixel. Comparing the MODIS product with field observations or high-resolution data at the site tends to induce errors due to spatial discrepancy. In this study, high-resolution data were first calibrated with the field observations. When evaluating the spatial representativeness of the MODIS product, Landsat pixels located at the center of the MODIS pixels were selected, so the analysis focused on this area could guarantee the spatial agreement to the greatest extent.

### 5.2. Difference Deduced by Upscaling Methods

When upscaling finer-resolution data to a coarser resolution, the simple average method is generally used. Mira et al. [10] assessed an equivalent PSF based on image correlation analysis using an aggregated albedo convolved with PSF over an agricultural landscape. The results indicated that convolving the PSF can reduce uncertainty by up to 0.02 (10%). We checked the difference deduced by the upscaling method.

#### 5.2.1. Simple Average Method

The TM value was first averaged then compared directly with the MODIS data. Figure 5 shows a comparison between the MODIS daily albedo data and the TM data. The results indicate that these datasets agreed well with RMSE less than 0.03 and $R^2$ greater than 0.92. Although the recommended aggregation scales were scale 2 and 3 (21 × 21 and 23 × 23 TM pixels), the difference in accuracy between the scales was not significant ($R^2$ ranged from 0.9341 to 0.9442; RMSE ranged from 0.0239 to 0.0249; the biases were nearly identical).

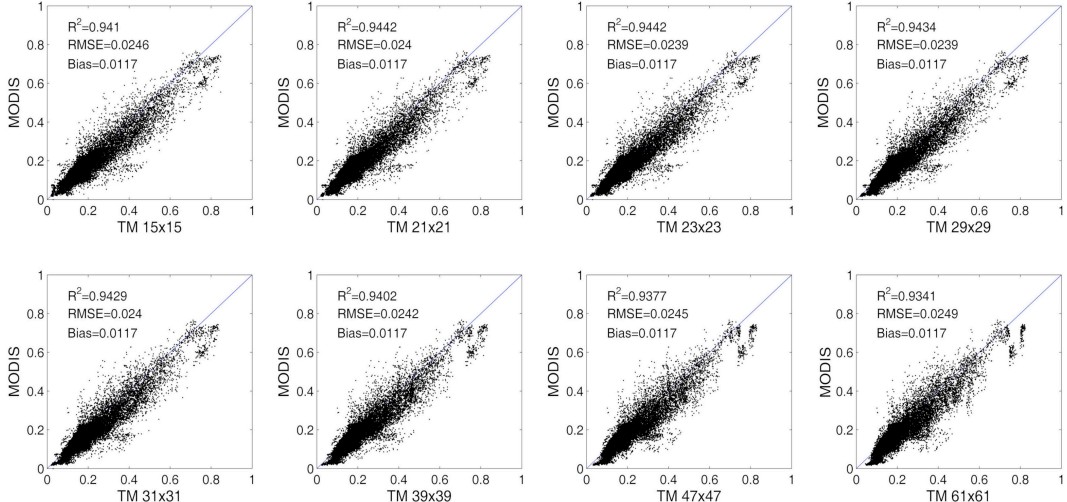

**Figure 5.** Comparison between MODIS albedo data with TM data aggregated at (**a**) 15 × 15, (**b**) 21 × 21, (**c**) 23 × 23, (**d**) 29 × 29, (**e**) 31 × 31, (**f**) 39 × 39, (**g**) 47 × 47, and (**h**) 61 × 61 scales.

#### 5.2.2. Aggregation with PSF

MODIS gridded products are the outputs of a sampled image system that combines an imaging system with a sampling procedure. The imaging system used was characterized by the sensor PSF, which is considered as the convolution of line spread functions in the along-scan and the along-track directions [10,40]. The FWHM is an important parameter in characterizing pixel resolution. We implemented the PSF function to TM data upscaling. The FWHM was set as the first eight scale values in Table 2, and a weight smaller than 20% was neglected [10].

Table 5 shows a comparison between MODIS data and the TM albedo aggregated with the PSF. The first aggregation scale (15 × 15 TM pixels) has a high $R^2$ value (0.944), while the second and third aggregation scales (21 × 21 and 23 × 23 TM pixels) has the lowest RMSE (0.0239).

Comparing the results of aggregation with the simple average and those obtained by convolving them with the PSF, the highest $R^2$ value appeared when the aggregation scale was 23 × 23 TM pixels. The RMSE of the simple average and those obtained through the convolution of the PSF were nearly identical (0.0239 and 0.0238, respectively), indicating that for all datasets, the difference deduced by the aggregation method was not significant.

**Table 5.** Comparative accuracy of MODIS albedo and TM data aggregated with PSF.

| Scale | 1 | 2 | 3 | 4 | 5 | 6 | 7 | 8 |
|---|---|---|---|---|---|---|---|---|
| TM pixels | $15 \times 15$ | $21 \times 21$ | $23 \times 23$ | $29 \times 29$ | $31 \times 31$ | $39 \times 39$ | $47 \times 47$ | $61 \times 61$ |
| $R^2$ | 0.944 | 0.9437 | 0.943 | 0.9404 | 0.9395 | 0.9354 | 0.9309 | 0.9207 |
| RMSE | 0.0239 | 0.0238 | 0.0238 | 0.024 | 0.0241 | 0.0243 | 0.0245 | 0.0248 |
| BIAS | 0.0117 | 0.0117 | 0.0116 | 0.0115 | 0.0114 | 0.011 | 0.0103 | 0.0081 |

## 5.3. Effect of Land Surface Heterogeneity

We then focused the analysis on tower located pixel for each site. For each scene, the semivariogram was calculated, from $15 \times 15$ to $77 \times 77$ TM pixels at a step of 2 TM pixels. Figure 6 shows the heterogeneous number of scenes at each step (sill value larger than 0.001). On average, 257 scenes were heterogeneous on 42 sites.

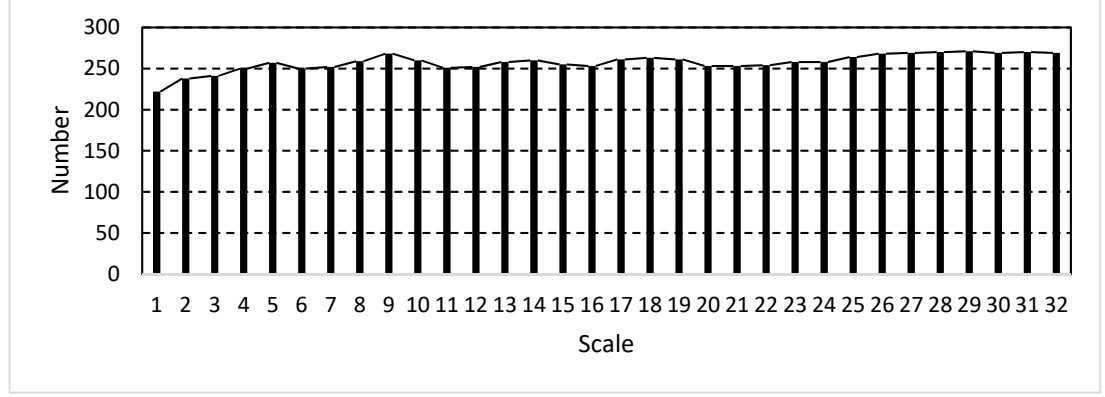

**Figure 6.** Number of heterogeneous scenes at each scale from $15 \times 15$ to $77 \times 77$ TM pixels at a step of 2 TM pixels.

We divided the heterogeneous landscape into three types. Figure 7 shows the general land surfaces and their spatial variations. Figure 7A shows a cropland from US-Ne3 site, located at the center of a rain-fed maize–soybean rotation field. The crop had been harvested, and only bare soil was explored in the scene. The trend in the variation in sill value was not significant with increasing scale. This meant that although the landscape was heterogeneous, the magnitude of heterogeneity at the scales was stable. The difference between the MODIS and the TM values decreased with increasing scale. In this case, with the scale enlarged, the influence of land surface heterogeneity diminished. The MODIS pixel represents an area much larger than its nominal resolution.

Figure 7B shows a US-Fmf site. This was an evergreen needleleaf forest site. The land surface heterogeneity was mainly due to the snow in the upper-right corner. With increasing scale, the land surface became more heterogeneous and the sill value increased from 0.0034 to 0.0059. The discrepancy between MODIS and aggregated TM pixels increased accordingly. For this situation, we may conclude that the more heterogeneous the land surface is, the greater the discrepancy is.

Figure 7C is CA-Fuf site. It is an evergreen needleleaf forest site not far from the US-Fmf site. The land surface heterogeneity was much higher than in the former two scenes, mainly due to the irregular surface and snow. The discrepancy between MODIS and TM albedo was correspondingly significant. In this case, we can hardly determine the effective spatial representativeness of the pixel.

Land surface heterogeneity is a key factor affecting effective spatial representativeness. In general, the more heterogeneous the land surface is, the larger the effective spatial representativeness is. When the magnitudes of land surface heterogeneity at different scales are similar, with the scale increased, land surface homogeneity increases.

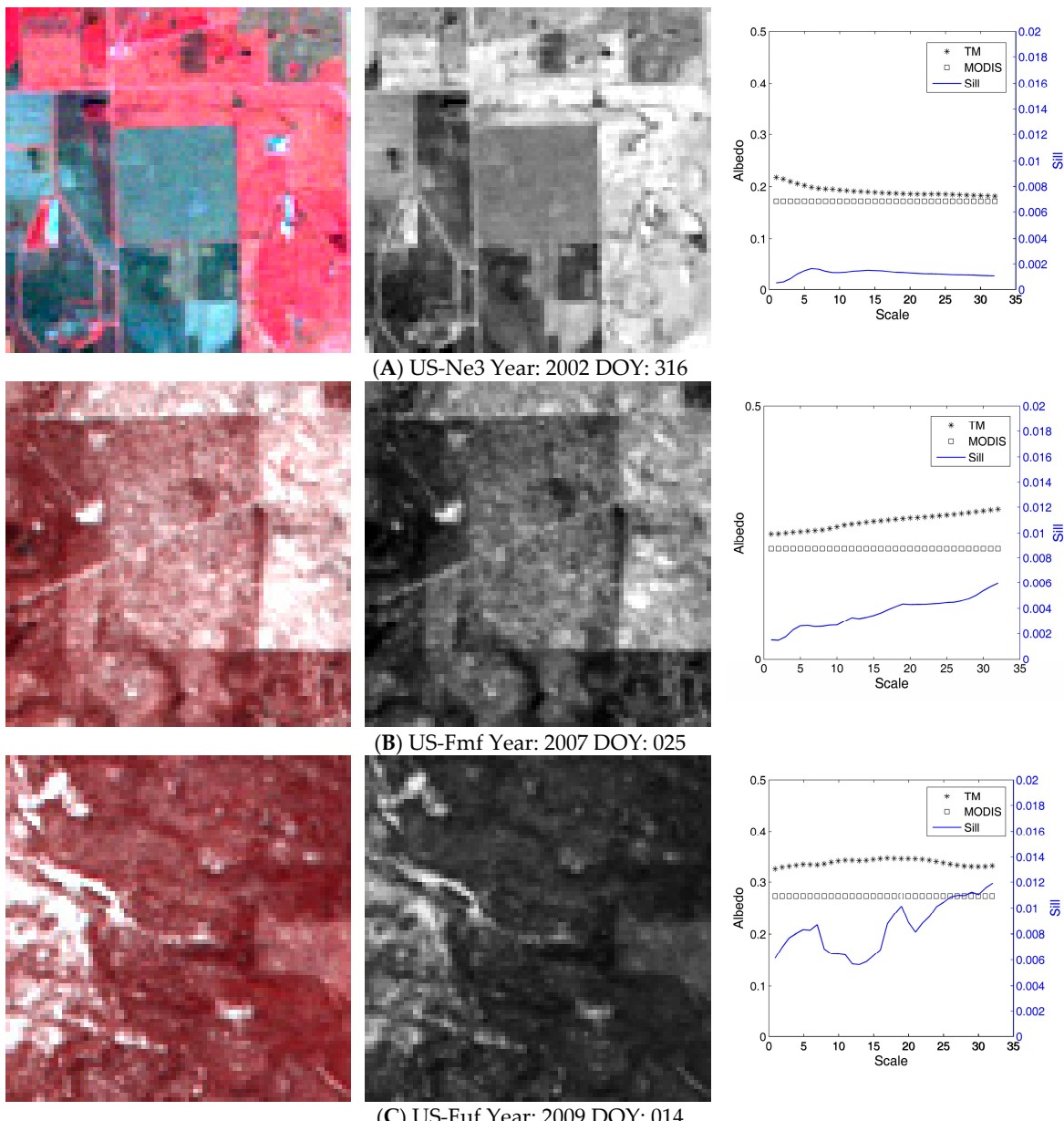

**Figure 7.** Three heterogeneous land surface types at (**A**) US-Ne3 cropland site, (**B**) US-Fmf evergreen needleleaf forest site and (**C**) CA-Fuf evergreen needleleaf forest site. For each site, the left figure is a false color composition of TM data, the middle figure is albedo from TM data, the right figure shows the aggregated TM albedo values at each scale, MODIS albedo, and sill value from scales 1 (15 × 15 TM pixels) to 32 (77 × 77 TM pixels) at a step of 2 TM pixels.

## 6. Conclusions

Land surface albedo is an important component of surface energy budgets. The validation of the satellite product is a precondition for its scientific use. Prior to evaluating the remote sensing products, identifying the spatial representativeness of the products is essential. It can help ground sampling point settings and match remote sensing data from multiple sources but it is difficult to determine the effective spatial representativeness of satellite albedo product from a physical perspective, as multi-temporal data are composed to derive the product.

In this study, we evaluated the spatial representativeness of MODIS daily albedo product (MCD43A3) at AmeriFlux sites. Among 1820 paired high-resolution (TM) and coarse-resolution (MODIS) albedo over different land cover types used in this study, around 74.5% pixels were found

to be heterogeneous pixels. In order to derive the most reliable spatial representativeness of MODIS albedo product, the land surface heterogeneity was first assessed by the field-calibrated TM albedo; semivarioagrams were then calculated from 30 m Landsat data at different spatial scales. Sill value and relative coefficient of variation were employed as key indices to determine the land surface heterogeneity. The 30 m Landsat albedo data was aggregated to 450 m–1800 m using direct average method and convolved with PSF method. The aggregated data was then compared with MODIS albedo product. The spatial representativeness of MODIS albedo product was determined according to the surface heterogeneity and the consistency of MODIS data and the aggregated TM value.

The results indicate that for most MODIS pixels their spatial representativeness tend to be larger than the 500 m nominal resolution. More specifically, for evergreen broadleaf forests, deciduous broadleaf forests, open shrublands, woody savannas, and grasslands, the effective spatial representativeness of the MODIS albedo was about 630 m; for mixed forest and croplands, the effective spatial resolution was about 690 m. The accuracy of the MODIS 500 m albedo product was high, with a correlation coefficient of 0.94 and RMSE 0.024 when compared with the calibrated TM albedo estimates. The choice of spatial aggregation method between simple spatial averaging and PSF-weighted averaging did not result in any significant difference in determining the spatial representativeness of MODIS albedo. It is also found that the spatial representativeness was difficult to determine at the sites where surface heterogeneity was very high (e.g., covered with evergreen needleleaf forest or partial snow).

In this study, long time period and large space data sets are used for spatial representativeness evaluation at 109 AmeriFlux sites with five land cover types when former works mainly focused on a specific research area. The availability of the 30 m Landsat albedo data set makes it possible for the analysis to be carried out at sites with different land cover types. There are many high-level remote sensing products, in this work, we only focus on evaluating the spatial representativeness of MODIS daily albedo product. Similar work is also worth for other products.

**Author Contributions:** H.Z. and S.L. conceived and designed the experiments. H.Z. performed the experiments and analyzed the data. J.W. and Y.B. helped the experiment and paper writing. T.H. and D.W. helped technology implementation of the albedo data processing. H.Z. wrote the paper. All the authors reviewed and provided valuable comments for the manuscript.

**Funding:** This research was supported by the National Natural Science Foundation of China under grants 41801242 and 41771379, the Key research and development program of China under grants 2016YFB0501404, 2016YFB0501502, the Chinese 973 Program under grant 2013CB733403.

**Conflicts of Interest:** The authors declare no conflict of interest.

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
