# Peer review of "Evaluating the Spatial Representativeness of the MODerate Resolution Image Spectroradiometer Albedo Product (MCD43) at AmeriFlux Sites"

_remotesensing, doi:10.3390/rs11050547_

Round 1
Reviewer 1 Report
This paper makes comparison between different levels of aggregation of a fine-resolution of a satellite-based surface albedo product, along with a coarse-resolution product for evaluation of the merits of each product. The paper has appropriate subject material for Remote Sensing. Its overall technical merit is sound. The writing and use of English is excellent with only a few exceptions.
Major comment:
My biggest concern is that you need to better explain how the in situ albedos at the AmeriFlux sites were used to calibrate the satellite products. Did you simply assume that the albedo was homogeneous at the 30 m scale of Landsat? (This might be a reasonable assumption, but you need to state it and give at least a cursory justification.)
Minor comments:
1. It’s strange that the family names of some authors are not capitalized. Is this a tradition that I am not aware of?
2. I believe the e-mail addresses of the co-authors located at the University of Maryland should not include “.cn”.
3. All acronyms need to be defined. In particular, lines 91-92 are dense with acronyms, mostly not defined, but also check the entire paper. TM is defined in the abstract, but should be defined again in the body (you did this for MODIS).
4. Line 63: I suggest writing out “tens of meters” instead of “10s of m”. My first thought was that “10s” meant ten seconds.
5. Dense small print is easier to read when broken into smaller paragraphs. In particular, I suggest you find a point to break the paragraph on lines 65-86 in two.
6. Lines 115-117: 1995 and 2013 are the earliest and latest years of data available from any of the sites, but this seems to say that all of those years are available at every site. I suggest “Radiation data were collected every half hour during the years indicated in Table 1 for each site. The overall range of years recorded at one site or more is 1995 to 2013.”
7. Line 119: Change “year” to “years” and “is” to “are”. (“Data” is a plural noun.)
8. Line 131: This acronym should be “LP DAAC” rather than “LP DACC”.
9. Line 146: There are multiple spaces before “He”.
10. Line 152: This algorithm should be called “nearest neighbor”. The next sentence repeats this one nearly exactly.
11. Lines 157-158: This is a sentence fragment (dependent clause) that extends across these two lines. Should it be connected by a comma with the following sentence?
12. Lines 184 and 203: There is extra space before “[47]”.
13. Lines 205-206: “RCV” should have “CV” subscripted.
14. Line 224: “I.e.” is for the Latin “id est”, meaning “that is”, and applies if you are going to restate something with exactly the same meaning. Here, you should use “e.g.” for “exempli gratia” meaning “for example”, since you give only one pair of aggregations out of several that exist.
15. Lines 228-229: This should say “between each pair of scales in table 2”, since your are using pairs and table 2 lists individual scales of aggregation.
16. Line 313: "2" should be superscripted in "R2".
17. Line 351: The meaning of “obtain” used here is much more rarely used than its other meaning, and seems kind of awkward. I suggest just saying “was”.
18. Line 357: Reverse the order of the words “correspondingly” and “significant”.
Author Response
This paper makes comparison between different levels of aggregation of a fine-resolution of a satellite-based surface albedo product, along with a coarse-resolution product for evaluation of the merits of each product. The paper has appropriate subject material for Remote Sensing. Its overall technical merit is sound. The writing and use of English is excellent with only a few exceptions.
Thanks you very much for your suggestions and comments. Those comments are all valuable and very helpful for revising and improving our paper. We have thought over all of the comments and suggestions. Revised portion are marked in blue in the paper. We sincerely hope all of the modifications meet your requirements.
Major comment:
My biggest concern is that you need to better explain how the in situ albedos at the AmeriFlux sites were used to calibrate the satellite products. Did you simply assume that the albedo was homogeneous at the 30 m scale of Landsat? (This might be a reasonable assumption, but you need to state it and give at least a cursory justification.)
Thanks for your comment! The field observation can cover the TM 30 m scale, so we think the land surface at 30 m scale is homogeneous enough and field observation can compare directly with TM data. We added the assumption in the text. And the most important thing when compare field observation with TM data is to guarantee the temporal and spatial consistence of the two data sets, so we averaged 3 × 3 TM pixels around the tower to guarantee the spatial consistence. Please find the revision in line 160-164.
Minor comments:
It’s strange that the family names of some authors are not capitalized. Is this a tradition that I am not aware of?
We are very sorry for the mistake! They should be capitalized. The authors names are corrected in line 5.
2. I believe the e-mail addresses of the co-authors located at the University of Maryland should not include “.cn”.
Thanks for your conscientiously work! We have correct them in the revised version. Please find them in line 11 and 12.
3. All acronyms need to be defined. In particular, lines 91-92 are dense with acronyms, mostly not defined, but also check the entire paper. TM is defined in the abstract, but should be defined again in the body (you did this for MODIS).
Thanks for your serious work! We added the full name for all the acronyms in the text in line 91-95. And checked the other acronyms through the paper.
4. Line 63: I suggest writing out “tens of meters” instead of “10s of m”. My first thought was that “10s” meant ten seconds.
The words have been corrected according to your suggestion to avoid misunderstanding in line 63.
5. Dense small print is easier to read when broken into smaller paragraphs. In particular, I suggest you find a point to break the paragraph on lines 65-86 in two.
The long paragraph is divided into two. The first paragraph focuses on introducing similar spatial representativeness assessment work and the second paragraph focuses on introducing the needs of evaluating the spatial representativeness of MODIS product. The first paragraph is from line 65 to 80, the second paragraph is from 81 to 86.
6. Lines 115-117: 1995 and 2013 are the earliest and latest years of data available from any of the sites, but this seems to say that all of those years are available at every site. I suggest “Radiation data were collected every half hour during the years indicated in Table 1 for each site. The overall range of years recorded at one site or more is 1995 to 2013.”
Thanks for your suggestion. We have rewritten the sentence according to your suggestion in line 117-119.
7. Line 119: Change “year” to “years” and “is” to “are”. (“Data” is a plural noun.)
Thanks for your serious work! The error has been corrected in line 120.
8. Line 131: This acronym should be “LP DAAC” rather than “LP DACC”.
Thanks for your serious work! We are very sorry for the mistake! The acronym has been corrected in line 133.
9. Line 146: There are multiple spaces before “He”.
The multiple space is deleted in line 150.
10. Line 152: This algorithm should be called “nearest neighbor”. The next sentence repeats this one nearly exactly.
Thank you for your reminding! The repeated sentence has been deleted and the nearest neighbor algorithm is used in line 155-156.
11. Lines 157-158: This is a sentence fragment (dependent clause) that extends across these two lines. Should it be connected by a comma with the following sentence?
There is a mistake in this sentence. It should be a comma instead of full stop. The sentence has been corrected in line 160.
12. Lines 184 and 203: There is extra space before “[47]”.
Thanks for your serious work! The extra space has been deleted in line 190 and 209.
13. Lines 205-206: “RCV” should have “CV” subscripted.
The RCV is used in line 212.
14. Line 224: “I.e.” is for the Latin “id est”, meaning “that is”, and applies if you are going to restate something with exactly the same meaning. Here, you should use “e.g.” for “exempli gratia” meaning “for example”, since you give only one pair of aggregations out of several that exist.
Thanks so much for your detailed explanation of the difference of these two abbreviations! The right word is used here in line 230.
15. Lines 228-229: This should say “between each pair of scales in table 2”, since your are using pairs and table 2 lists individual scales of aggregation.
The original express is inappropriate and vague. The sentence is corrected according to your suggestion in line 234-235.
16. Line 313: "2" should be superscripted in "R2".
R2 is used in line 294 and 319.
17. Line 351: The meaning of “obtain” used here is much more rarely used than its other meaning, and seems kind of awkward. I suggest just saying “was”.
Thanks for your suggestion! Word “was” is used here in line 357.
18. Line 357: Reverse the order of the words “correspondingly” and “significant”.
The words “correspondingly” and “significant” has been reversed in line 363.
Reviewer 2 Report
The issue dealt with by the authors is timely. The only negative feature of the manuscript: the amount of text the reader must pass through is disproportionately large compared to the strength of the conclusion. It should be clearly written in the conclusions, what practical significance can the results of the authors' research have.
The letter abbreviation RMSE is explained in line 164, and should already be in line 93.
At the axes of the graphs in Figure 7 should be used greater letters. Numbers on the right are invisible (truncated).
I would prefer that there were no dots in mathematical expressions as a sign of multiplication, because the dot commonly means scalar multiplication of vectors.
Author Response
The issue dealt with by the authors is timely. The only negative feature of the manuscript: the amount of text the reader must pass through is disproportionately large compared to the strength of the conclusion. It should be clearly written in the conclusions, what practical significance can the results of the authors' research have.
Thank you very much for your suggestions and comments. We enriched the conclusions section. Four parts are included: 1) why we need to evaluate the spatial representativeness of the remote sensing product. 2) how we evaluated the spatial representativeness of MODIS daily albedo product with TM data. 3) the main conclusion of the analysis: what is the proper spatial representative scale of the MODIS product. 4) the advantage/disadvantage of the study and further work needed. Please find the revised part in line 380-405.
The letter abbreviation RMSE is explained in line 164, and should already be in line 93.
Thanks for your serious work! The full explanation of RMSE has been moved to line 95.
At the axes of the graphs in Figure 7 should be used greater letters. Numbers on the right are invisible (truncated).
Greater letters are used in Figure 7 according to your advice. Please find the revised figure in line 368.
I would prefer that there were no dots in mathematical expressions as a sign of multiplication, because the dot commonly means scalar multiplication of vectors.
Thanks for your suggestion! We corrected formula 1 according to your advice in line 192.